# Understanding networks in low-and middle-income countries' health systems: A scoping review

Katherine Kalaris[1]*, Geoff Wong[2], Mike English[3,4]

1 University of Oxford, Oxford, United Kingdom, 2 Nuffield Department of Primary Care Health Sciences, University of Oxford, Oxford, United Kingdom, 3 Nuffield Department of Medicine, University of Oxford, Oxford, United Kingdom, 4 KEMRI Wellcome Trust Research Programme, Nairobi, Kenya

☯ These authors contributed equally to this work.
* katherine.kalaris@kellogg.ox.ac.uk

**Data Availability Statement:** The data is accessible on the Open Science Framework doi: 10.17605/OSF.IO/DJTMW.

**Funding:** ME is funded by a Senior Research Fellowship from the Wellcome Trust (#207522).

## Abstract

Networks are an often-employed approach to improve problems of poor service delivery and quality of care in sub-optimally functioning health systems. There are many types of health system networks reported in the literature and despite differences, there are identifiable common characteristics, uses, purposes, and stakeholders. This scoping review systematically searched the literature on networks in health systems to map the different types of networks to develop an understanding of what they are, when and what they are used for, and the purposes they intend to achieve. Peer-reviewed literature was systematically searched from six databases (Medline (Ovid), EMBASE (Ovid), Global Health (Ovid), the Cochrane Library, Web of Science Core Collection, Global Index Medicus's Africa Index Medicus) and grey literature was purposively searched. Data from the selected literature on network definitions, characteristics, stakeholders, uses, and purposes were charted. Drawing on existing frameworks and refining with the selected literature, a five-component framework (form and structure, governance and leadership, mode of functioning, resources, and communication), broadly characterizing a network, is proposed. The framework and mapping of uses, purposes, and stakeholders is a first step towards further understanding what networks are, when and what they are used for, and the purposes they intend to achieve in health systems.

## Introduction

Health systems are complex systems that may not function optimally due to high levels of demand, limited resources, structural inertia, insufficient financing mechanisms, and weak governance [1, 2]. Clinicians and health systems managers, among other health sector stakeholders, look to approaches to improve service delivery, quality of care, and clinical outcomes that are within their capabilities and available resources. Health system networks are an oft employed approach to solve these problems [3].

The funder had no role in study design, data collection and analysis, decision to publish, or preparation of the manuscript.

**Competing interests:** I have read the journal's policy and the authors of this manuscript have the following competing interests: KK is a consultant for the World Health Organization on networks of care for maternal and newborn health. ME established the Clinical Information Network in Kenya and receives funding for this research platform.

Many different types of networks exist in the field of healthcare, ranging from global-level advocacy networks to community-based networks. We distinguish these from the common representation of many public sector health systems organized in tiers with service delivery units formally linked through geographic or hierarchical networks. Our interest is in the networks that form when groups of health system actors from across level and sectors of care, entities, and geographies come together in a distinct way and work together with the aim to improve service delivery, quality of care, and/or health system functioning. These networks are not a parallel structure to the public sector health system but often represent a new layer on top of or within the existing health system, thereby potentially reinforcing it. In this paper it is these supplementary system strengthening networks that are the subject of interest (referred to as simply networks hereafter).

There are various network definitions, typologies, and frameworks in the literature but there is "no single, consensually agreed definition of what constitutes a 'network'" [4]. Box 1 provides examples of published network definitions. Networks in health systems are often clinically and programmatically focused, working to achieve a range of purposes, including to address variations in practice and outcomes [5], improve processes and quality of care [6], increase evidence-based practice [7], facilitate change [8], share knowledge and innovation [4], and achieve joint goals [9]. Despite this diversity in purpose across networks, networks have identifiable similar characteristics, such as, having a shared communal goal or vision, strong visionary leadership, robust communication, and trust [9–12]. Networks have been established in health systems in many countries, for example, networks of safety to improve maternal and newborn care and survival in the Himalayas of Nepal [13], public-private sector networks for maternal and newborn care in Tanzania and the Philippines [14, 15], quality improvement networks in India and the US [16, 17], and clinical networks in Australia and Canada [12, 18, 19]. However, reports on networks in low-and-middle income countries (LMICs) are infrequent as is research that looks across network types to understand what they are and why they are used in practice.

> ## Box 1. Examples of definitions of networks [16, 20–23]
>
> **Clinical network:** "voluntary clinician groupings that aim to improve clinical care and service delivery using a collegial approach to identify and implement a range of strategies across institutional and professional boundaries" (Brown et al 2016)
>
> **Networks of care:** "group of public and/or private health service delivery sites deliberately interconnected through an administrative and clinical management model which promotes a structure and culture that prioritizes client-centred, effective, efficient operation and collaborative learning, enabling providers across all levels of care, not excluding the community, to work in teams and share responsibility for health outcomes" (Carmone et al 2020)
>
> **Managed care network:** "Linked groups of health professionals and organisations from primary, secondary and tertiary care working together in a coordinated manner, unconstrained by existing professional (and organisational) boundaries to ensure equitable provision of high quality effective services" (Addicott & Ferlie 2007)
>
> **Health services network:** "the integration of diagnostic, therapeutic and care activities provided by different professionals and different organizations, in the hospital and in the community, that cooperate to achieve a shared mission" (Aspromonte et al 2017)

> **Quality improvement collaborative:** "a group of professionals from a single or multiple organization who get together to learn from one another, support and motivate each other in a structured approach with the intent of improving quality of health services" (Murki et al 2018)

In this paper we looked at networks that were distinctly established, whether informally or formally, from the top-down or bottom-up to complement existing health system structures. These networks were focused on service delivery, quality improvement, learning, and health systems functioning.

## Rational

The literature on networks in health systems is diverse and there is currently no overarching mapping of the existing evidence or a common description of what makes up health system networks. As networks become a more frequent approach to improve health systems' functioning, service delivery, quality of care, and clinical outcomes, it is important to understand the key components that facilitate their work towards achieving these changes. This common understanding can inform future efforts to develop and scale networks in health systems by enabling a more targeted consideration to the key aspects of a network that need to be put in place.

Scoping reviews are a particularly useful method when the body of literature, is of a "large, complex, or heterogeneous nature not amenable to a more precise systematic review" [24]. Networks are a complex approach integrated in a complex system (a health system) and scoping reviews can help to "clarify a complex concept and refine subsequent research inquiries" [25]. It is a useful approach for mapping, summarizing, and identifying gaps in the literature of complex concepts [26]. There have been few published reviews on networks in health systems. An initial search of the published literature identified one scoping study on networks of care drawing on high-income country (HIC) and LMIC literature, a systematic review on clinical networks in HICs, and a systematic review on quality improvement collaboratives which included mainly studies from HICs [10, 11, 21]. These three reviews focused on only one type of network each and therefore a new scoping review was determined to be a useful approach to map and summarize the current literature and expand what is known by looking at the different types of health system networks with a specific focus on LMIC literature. Furthermore, this scoping review is a precursor to a realist review [27] and evaluation focused on understanding how and why networks work to change practices in LMIC health systems.

### Aim, objectives, and research question

This scoping review, by looking at the existing research on the different types of networks in health systems, sets an initial field of inquiry on networks in LMIC health systems. The objectives of the review were to: 1) perform a systematic search of published and grey literature on networks in health systems; 2) map and summarize network types, definitions, stakeholders, characteristics, uses, and purposes, and other key findings relevant to networks in health systems; 3) identify gaps in the literature related to networks in LMIC health systems; and 4) propose an operational typology of networks in health systems. The research question guiding this scoping review was: What is a network, when are they used, what are they used for, and what purposes are they intended to achieve in health systems?

## Methods

This scoping review was conducted in accordance with the original five phase scoping study framework developed by Arksey and O'Malley (2005) and took into consideration additional recommendations on scoping reviews from Levac et al. (2010) and Peters et al. (2015) [24–26]. A protocol was developed and registered with the Open Sciences Framework on 8th February 2021(https://osf.io/8bg79/registrations). The process and results are reported according to the PRISMA Extension for Scoping Reviews [28].

### Eligibility criteria

The literature search process focused on health system networks. Networks were defined for the eligibility criteria as groups of facilities and/or healthcare affiliated stakeholders linked formally or informally, horizontally or vertically, through programs, interventions, activities, or initiatives. The development of the eligibility criteria for the network concept was an iterative process; as the research team became more familiar with the literature, the eligibility criteria were refined to identify the most relevant literature. During the screening process the exclusion criteria were further expanded as it became clearer what would be relevant to answer the research question. Included types of networks were mostly clinically or programmatically focused, excluding, for example, networks that were purely research or academic networks, advocacy networks, disaster management networks, database or registry networks, laboratory or diagnostics networks, and family or home care networks. Updates to the eligibility criteria were discussed among the research team and applied to all retrieved literature. The complete eligibility criteria are available in S1 Table.

Literature from HICs and LMICs, as classified by the World Bank for the 2021 fiscal year [29], and published between 2000 to 2021 was eligible for inclusion. While the focus of this review is on networks in LMICs, limited research focused on these geographies has been published and so HIC literature that was inadvertently returned during the search process or that the authors were already aware of was screened for eligibility and included if relevant to the LMIC literature. The search process was open to peer-reviewed and grey literature in all languages but those published in English and French were prioritized for review due to research team capabilities.

### Information sources

Five databases were searched for relevant literature on 3rd February 2021: Medline (Ovid), EMBASE (Ovid), Global Health (Ovid), the Cochrane Library, Web of Science Core Collection. Global Index Medicus's Africa Index Medicus (AIM) was searched on 4th February 2021; only the AIM database was searched from Global Index Medicus due to database capabilities to process the search strategy as well as because it is the region of focus for subsequent research. These databases were selected with the aim of searching with both breadth and depth. Searching several different databases allowed us to cast a wide net of journals that might publish literature on networks in health systems. The Global Health and AIM databases were selected as the review focused on networks in LMICs and these databases were thought to potentially have a great number of LMIC examples. Grey literature was searched from the websites of bi and multilateral organizations, non-governmental organizations, and global initiatives. Additional details on the information sources are included in the Results Section. A modified update search was run on Medline, EMBASE, Global Health, and the Web of Science Core Collection on 27th April 2022.

## Search

The search strategy was iteratively developed through four steps outlined in Fig 1. The final search strategy is provided in S1 Appendix. The search strategy returned both peer-reviewed and grey literature and additional grey literature was purposively collected. Most of the grey literature provided additional examples of LMIC networks not published in peer-reviewed journals.

## Selection of sources of evidence

Titles and abstracts were screened against the eligibility criteria using the web application Rayyan [30]. The full text of selected titles and abstracts was assessed against the eligibility criteria. A second reviewer screened and reviewed a random sample of 15% of the titles and abstracts and of the literature selected for full text review. At the completion of screening and full text review, discordant decisions within the random sample were reconciled between the reviewers through discussion. Grey literature was reviewed for inclusion according to the same criteria.

## Data charting process and data items

The primary and second reviewer (for a 15% sample of included literature) charted relevant data and insights from the selected literature using the data charting instrument in S2 Table. Data charting by the second reviewer was to check that the charting process was applied by the primary reviewer in a consistent manner. This tool facilitated the charting of key information from the included literature and findings and insights relevant to the review question. Data items are defined in Table 1.

## Synthesis of results

Following the careful appraisal of the available literature, it was apparent that defining distinctly different types of networks and creating a typology as planned would not be feasible because of the diversity of networks reported in the literature meeting the eligibility criteria. Instead, to guide data synthesis, network typologies and frameworks on different types of networks (managed networks, research/academic network [31], clinical networks [10, 20], inter-organizational networks [9], public sector networks [32, 33]) were identified from the selected literature and through snowballing. These typologies and frameworks, presented in S3 Table, were compared for similarities. Four common network components were identified from these frameworks and typologies and compiled into a new draft framework.

To understand how the identified network components are operationalized in practice, charted data were mapped onto each of the four components of the draft framework and similar data were grouped together under each component; these data groups were labelled as practical characteristics. The frequency of occurrence in the literature of each practical characteristic was noted. Throughout this analysis, the authors remained open to emerging components, resulting in a final framework with five components that was simple, high-level, and flexible enough to accommodate many practical characteristics and applicable to the

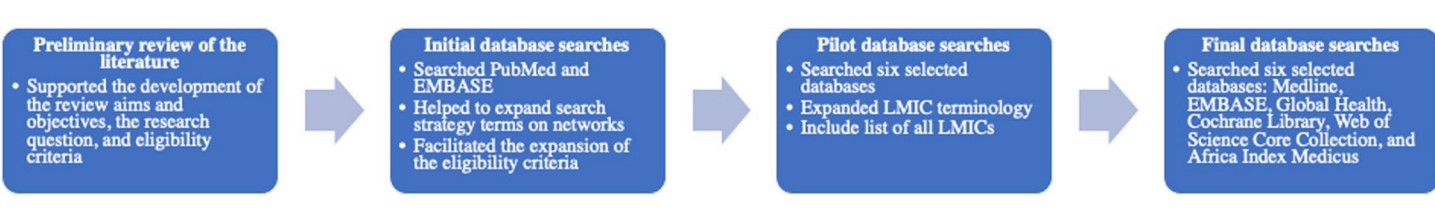

**Fig 1. Search strategy development steps.**

**Table 1. Data charting categories and definitions.**

| Category | Definition |
|---|---|
| Source citation | Publication bibliographical information |
| Intervention country | Country where the study took place or the focus of the publication |
| Aim/purpose | Goal or objective of the study, review, or publication |
| Methodology | Methodology of the study, review, or other type publication |
| Intervention | Description of the project, program, or activities that were the focus of the study, review, or other type of publication |
| Outcome/findings | Results, findings, or key insights from the study, review, or other type of publication |
| Stakeholders | Key people, roles, committees, entities, or structures involved in the intervention |
| Type of network | Name of the network as reported by the authors |
| Network definition | Definition of the network as stated by the authors |
| Network characteristics | Key descriptors, elements, and activities of the network |
| Network use | What the network does |
| Network purpose | Why was the network created or what was the overall goal |
| Other key findings or data relevant to the review questions | Potentially relevant or important information not included in another category |

range of networks in health systems. This process is outlined in Fig 2 and the final framework is presented in the results section.

In a distinct but complementary approach to answer the remainder of the research question on the uses and purposes of networks, the charted data on network use and purpose was mapped, similar data was grouped, and the frequency within each data group reported. These groupings of mapped data were then synthesized into overarching categories of uses and purposes. During the data charting, the authors decided to include network stakeholders as an additional data item. Stakeholder data was charted and synthesized following the same process.

## Results

### Selection of sources of evidence

The final database searches obtained 11,142 results, of which, 106 peer-review and 21 grey pieces of literature met the eligibility criteria. The update searches identified two additional pieces of literature that met the eligibility criteria. Fig 3 provides an overview of the evidence selection process.

### Characteristics of sources of evidence

Characteristics of the selected literature are detailed in S2 Appendix. The selected literature employed many different methods, including descriptive project overviews (32), quantitative

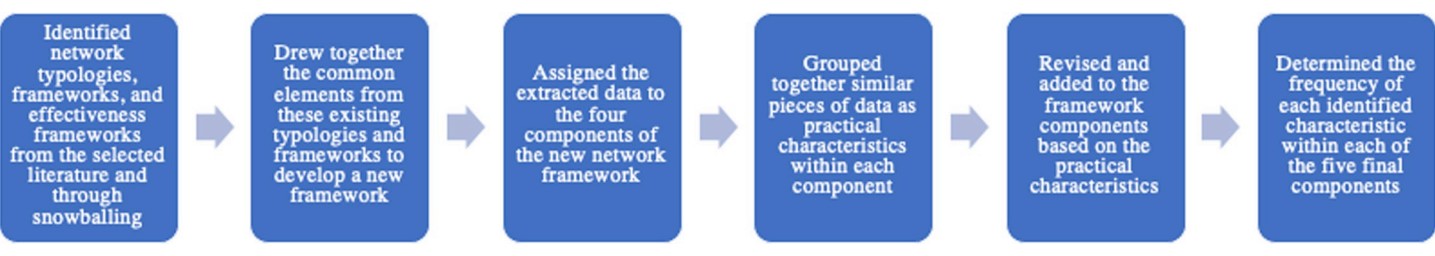

**Fig 2. Process of synthesizing the charted data.**

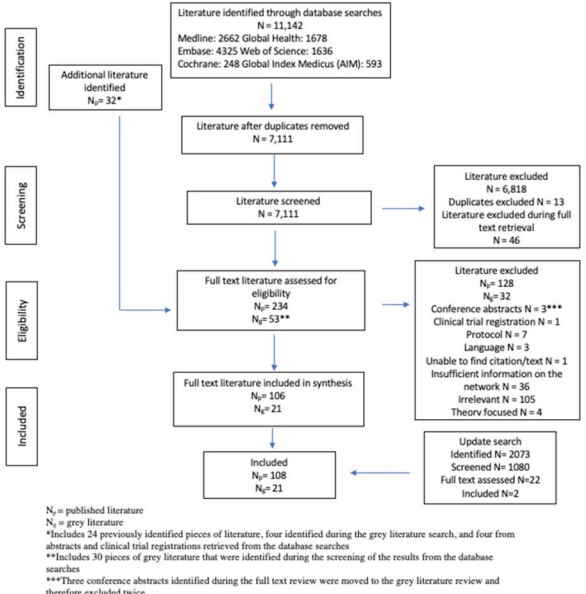

**Fig 3. Evidence selection process.**

studies (17), case studies (15), qualitative studies (15), reviews (12), and mixed methods (6) in the peer-reviewed literature and reports (7), briefs (7), manuals (4), and working papers (3) in the grey literature.

Over half (57%) of the selected literature focused on LMIC networks; this was expected based on the targeted search strategy. The Africa region represented most of the literature with 36 discrete inclusions of African countries. Fig 4 shows the literature breakdown by geography and literature type.

## Results of individual sources of evidence

The framework, presented below, helps to describe the overarching components that make up a network. Based on the literature, health system networks can be described by five common

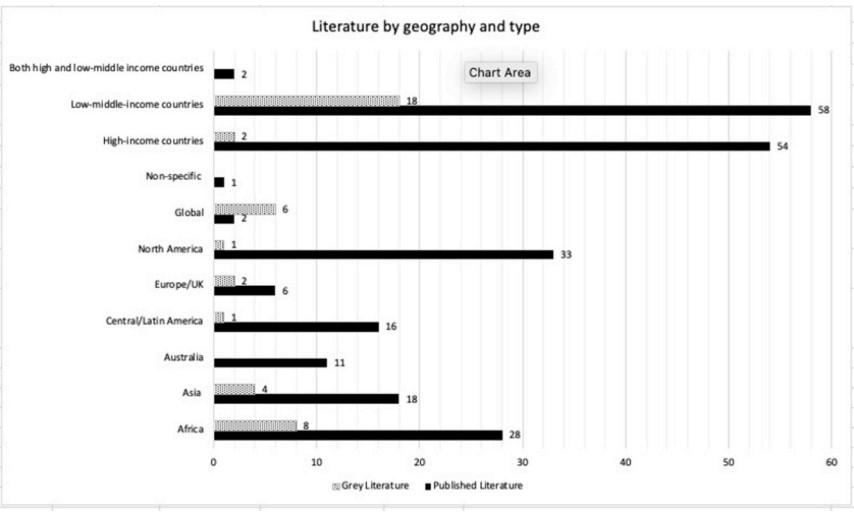

**Fig 4. Literature by geography and type included in the peer-reviewed and grey literature.**

components: form and structure, leadership and governance, mode of functioning, resources, and communication. The components are supported and illustrated by operational examples from the charted data on network characteristics. Mapping reported network uses and purposes from the literature answered when and what networks are used for and what purposes they intended to achieve in health systems. Network stakeholders were also mapped. S3 and S4 Appendices link the network components, practical characteristics, uses, purposes, and stakeholders to the individual sources of evidence.

**Components and characteristics.**   Based on a systematic literature search and the richness of the charted data, none of the identified typologies or frameworks were sufficient to describe a network and its components in a comprehensive manner. This was because they were focused on specific types of networks, each with some but not all of what seemed to be the essential network elements, based on the reviewed literature. Four key common components were identified from the existing typologies and frameworks: form and structure, governance and leadership, mode of functioning, and resources. Communication emerged as a new key component when data was mapped against the four components. Component labels were derived from existing typologies and frameworks or were key words in the literature. Each component is supported by practical characteristics that provide operational examples illustrating how the component manifests in networks in practice. Descriptions of the components are summarized in Table 2. Table 3 lists the practical characteristics by component with the number of references supporting each practical characteristic. S4 Table details the most frequent practical characteristics of each network component supported by examples from the literature.

## Form and structure

How a network is *formed and structured* has implications for how a network is governed and led, its management and clinical functions, how network members communicate, and the key

**Table 2. Network component descriptions.**

| Component | Description |
|---|---|
| Form and structure | The form of a network explains how the network was created. In a top-down or mandated approach, a government or clinical structure projects the network out to potential members. A bottom-up or organically developed network emerges from its members based on identified needs in their context. The network structure explains the linkages, horizontal or vertical, between different network members or stakeholders, including clinical entities, providers, or health system administrators and managers. |
| Governance and leadership | Network governance refers to how the network is managed and administered, and decision-making processes employed to manage the network. The network leadership includes the level and form of leadership needed to effectively manage the network. Level of leadership refers to type, seniority, and experience of the leader. Form of the leadership refers to the structure of the leadership team, for example a hybrid clinical and management leadership team or a network management committee. |
| Mode of Functioning | A network enacts its purpose through management and clinical functions. Management of a network includes monitoring compliance and accountability, while the clinical functions refer to the implementation and operationalization of policies and programs and the coordination of clinical care. |
| Resources | Based on the existing typologies and frameworks, the two main types of resources available to networks are human resources and financial resources. These take different forms depending on the context and purpose of the network. This component was expanded during the analysis to include information technology, commodities, and equipment. |
| Communication | A fifth component, communication, was added during analysis. Communication encompasses communication infrastructure as well as modalities and strategies. This component interlinks the four other components. |

**Table 3. Network components with corresponding practical characteristics.** The number of references in the selected literature are shown for each characteristic.

| Form and Structure • Creation of the network: top-down/mandated vs. bottom-up/organic • Linkages between entities | | Governance and Leadership • Leadership level • Leadership form • Management of network •Governance/decision-making processes | | Mode of Functioning • Management: monitoring compliance, accountability • Clinical: policy/program operationalization, care coordination | | Resources • Human resources • Financial | | Communication | |
|---|---|---|---|---|---|---|---|---|---|
| connecting across levels of the health system and entities | 42 | network meetings | 37 | knowledge / information sharing / education / learning | 67 | human resources | 43 | between network members | 24 |
| established vision, mission, shared values, targets, rules, roles, responsibilities, culture | 35 | network leadership / management | 31 | guideline / standards / protocols uptake / adherence | 57 | IT | 38 | communication infrastructure | 12 |
| network agreements / network mapping | 30 | working /sub-groups / task forces | 23 | data collection, analysis, use, quality | 56 | commodities/equipment | 14 | strengthening communication | 5 |
| partnerships / links to external stakeholders | 29 | government leadership / oversight | 21 | quality improvement | 51 | funding | 13 | effective strategies | 4 |
| linkages / engagement / alignment with government | 20 | network manager / coordinator / facilitator | 18 | care pathways / models of service delivery promotion / implementation | 47 | government funding | 12 | | |
| trust | 17 | local / clinical champions | 16 | training and/or supervision | 45 | supportive policies | 12 | | |
| multidisciplinary | 16 | steering / coordinating committees / groups | 15 | feedback—performance / performance management / care processes | 35 | transport | 11 | | |
| horizontal / lateral network | 15 | core leadership committee / team | 13 | work plans / strategic planning / project development | 25 | infrastructure | 9 | | |
| relationships / links between teams / internal | 15 | clinical leader—network coordinator / voluntary co-chairs | 12 | revise / standardize patient forms / records systems / tools | 24 | cost reduction / savings | 9 | | |
| link community to network | 11 | decision-making | 12 | mentoring / coaching | 24 | financial incentives | 8 | | |
| bottom-up / local / organic / informal | 7 | interdisciplinary / effective / open / multi governance | 12 | coordination | 23 | administrative / operational support | 8 | | |
| standard / formalized organizational structure | 7 | hybrid leadership—clinical (general—specialisms), program/operational, executive | 10 | changes in practice or service delivery | 19 | financial subsidies / free services | 5 | | |
| patient / family / consumer engagement | 6 | community / local leadership / ownership in activities | 9 | monitoring | 17 | UHC / insurance | 4 | | |
| vertical structures | 6 | focal point mid-level managers—clinical leaders | 8 | reporting | 16 | pay-for-performance / results | 4 | | |
| created on existing relationships | 6 | executive support / strategic and technical assistance | 7 | assessment / evaluation | 14 | financial management support | 4 | | |
| voluntary clinician / hospital involvement | 6 | governance structure | 6 | collaboration | 14 | resource sharing | 3 | | |
| combination of top-down and bottom-up | 6 | stakeholder management | 6 | accountability | 10 | service payments | 2 | | |
| mandated / top down on policy and strategic direction | 4 | administrative core / support | 3 | performance comparison / benchmarking | 10 | non-financial incentives | 2 | | |
| coordination body | 2 | | | quality of care | 9 | | | | |
| peer to peer network | 3 | | | role clarification / orientation of new providers on preferred practices | 8 | | | | |
| multi-organizational | 2 | | | MPDSR | 7 | | | | |
| open membership | 2 | | | audit and feedback—data | 5 | | | | |
| supra-network | 1 | | | leadership training / enhance skills / relationship building | 4 | | | | |

*(Continued)*

**Table 3.** (Continued)

|  |  | clinical audit | 4 |  |  |  |
| --- | --- | --- | --- | --- | --- | --- |
|  |  | teamwork | 4 |  |  |  |
|  |  | shared responsibility | 2 |  |  |  |
|  |  | mentorship and training—management | 1 |  |  |  |

resources it needs to function. While networks are formed in different ways and have various structures, in the HIC literature, networks developed from the bottom-up, for example clinician led networks in Australia, or with a mixed top-down/bottom-up approach may be more successful than those formed by a top-down approach, such as managed clinical networks in the UK [10, 12, 22]. The process of creating linkages across different levels and sectors of the health system and how these link actors illustrates how networks are structured and form [3, 34]. For example, a maternal-neonatal health network in Indonesia linked public and private referral hospitals and community health centers and providers. To create the network, facility staff, district officials, and civil society mapped the most efficient referral pathways which were reinforced with a communications system. Connections were created beyond the referral pathways by linking providers at different levels of facilities through mentoring teams. Health facility staff were connected to district officials through district health system data working groups. The linkages extended to the community level with maternal and child health motivators working to address issues related to maternal and newborn survival. These linkages and relationships generated trust and a shared vision among network actors, increased motivation and performance, and were a contributing factor to results achieved by the network [35].

## Governance and leadership

The governance and leadership of networks play an important role in the establishment and ongoing functioning of a network. Networks are *governed and led* through various mechanisms depending on structure, use, purpose, and members. One common way networks are governed is through network meetings, which vary in type, frequency, and participation, to support network management and administration. Network meetings differ in approach, but most are a means to collaborate, coordinate, and share information. Horizontally structured networks are more likely to employ meetings as a way of coordination and collaboration; for example the Clinical Information Network (CIN) in Kenya links county level hospitals around a central network coordinator and holds regular meetings for network members to exchange and learn from each other and promotes local midlevel clinical management [36]. A hybrid horizontal–vertically structured maternal-neonatal health network in Ecuador employed different network meetings to implement network activities. The network held meetings to engage local government representatives on national priorities and laws, monthly meetings between Traditional Birth Attendants, Community Health Workers, and health center staff to review the status of pregnant women in the community and to address potential barriers, and monthly quality improvement team meetings focused on improving access and quality of services and to review and resolve issue at the community and hospital levels. These meetings aimed to engage network actors from the community and different levels of clinical and health system management in the implementation and running of the network. This helped to transform fragmented healthcare services into a "coordinated, cooperative entity" [37].

Network leadership takes different forms. In vertical or mandated networks leadership is more centralized, which can lead to power struggles in decision making and implementation of

network activities between the central network administrators and clinician network members [22]. Horizontal networks are more likely to have distributed leadership. A quality improvement collaborative in the US was formed of multiple layers of leadership: a leadership team responsible for network project infrastructure, a steering committee to develop structure for projects and coordinate project selection, a project development team to provide clinical guidance on project design, a project management team to track progress and organize monthly meetings and learning sessions, and facility advisors to provide oversight and feedback to sites [38].

## Mode of functioning

A network's *mode of functioning* describes what the network does to try to achieve its purpose, for example through knowledge and information sharing, education, and learning activities. Quality improvement initiatives are common examples in the literature of using knowledge and information sharing and learning activities to work towards goals. A quality improvement collaborative in Tanzania, aiming to improve provider-initiated HIV testing and counselling and linkages to care and treatment, held regular learning sessions for the participating sites to share progress toward aims, change ideas, plan–do–study–act cycles, lesson learned, and best practices. The way in which a network functions enables the implementation of "locally appropriate process-focused change interventions that lead to sustained system improvements" [39]. Networks will also develop or revise guidelines, standards, and protocols and promote their uptake and adherence as a way to work towards achieving their purpose. For example, in Kenya's CIN network members were engaged in the development of national clinical guidelines and promoted and distributed them throughout the network. Their championed use of guidelines trickled down to use by junior clinicians and broke down previous barriers hindering guideline uptake [40].

## Resources

Networks need different types of resources to function, mainly human resources, IT, commodities and equipment, funding, and supportive policies. In HIC literature, different aspects of *human resources* were cited as one of the key elements that made a network effective or successful, such as sufficient staffing, engaged network members, or strong network leadership [10, 12, 20]. A network can provide staff with additional roles, in the case of clinicians acting voluntarily as network co-chairs in clinical networks in Australia [20] or they may need to recruit a dedicated network coordinator [6]. Human resources are a key element in LMIC networks as well; for example, networks in Nepal, Tanzania, and Madagascar filled gaps in clinical staffing by working with local government to support recruitment and short-term salaries [13, 14, 41]. Human resources make things happen in networks and work towards achieving their purpose. In Kenya's CIN, the county hospitals' mid-level managers act as the network's focal points and play "boundary spanning roles" with clinical and management responsibilities. The success of the network lays with these focal points and potentially their ability to build skills in leading multidisciplinary teams [42].

## Communication

The successful communication between network members contributes to reaching network aims. In HIC studies of clinical and service delivery networks, communication is highlighted as a key to network success [10, 18, 43]. The importance of this concept is also exhibited in the LMIC literature. In Metro-Manila, Philippines, a network linking a tertiary public hospital and public and private midwifery clinics facilitated communication between levels and sectors of care, enabling the transfer of information on referred patients and unit capacity as well as sharing updates. A dedicated phone line for the network is carried by the Obstetrics and Gynecology clinician on-call and chat groups of network participants were set up. Communication

between network members helped facilitate the growth of trusting relationships between different cadres, sectors, and levels of care, which enabled the timelier transfer of pregnant women from midwifery clinics to more complex care and the out-bound referral of low-risk cases from the crowded tertiary hospital [15].

**Network uses and purposes.** There is significant overlap in the results on reported network uses and purposes and it was challenging to differentiate between them due to the way networks are reported in the literature. Network uses were mapped to 32 concepts and network purposes to 19 concepts. These groups of concepts were summarized into five overarching categories: improving and providing quality care and services, improving the health system, improving provider capacity, improving patient use and experience, and improving outcomes (purpose only). Categories were labeled to represent the similar concepts contained within each category. The largest category, improving and providing quality care and services, contained concepts related to the offering and coordination of quality care, improving access to care, and quality improvement. Concepts around changes to aspects of the health system, facilitating referral, and fostering partnerships composed the category improving the health system. Improving provider capacity contained concepts specific to changes in clinical practice, the uptake of standard guidelines, and building capacity and skills. The improvement of patient use and experience represented concepts mainly around the increased uptake of services and sharing of information between providers, patients, and caregivers. Lastly, the category improving outcomes focused on improvement in clinical outcomes. Tables 4 and 5 list the uses and purposes of networks by category, respectively, from the reported literature. S4 Appendix contains the reported uses and purposes by category linked to specific literature references.

The following examples highlight some of the network uses and purposes. In Ecuador, a multi-level maternal and newborn health service delivery network aimed to "improve access to and quality of EONC across the care continuum" [37]. Shared-care networks for child cancer care in Ghana and Bangladesh worked to coordinate care to enable patients to access care closer to their homes [44]. Networks often aim to foster changes in practice, for example the CIN in Kenya works to improve use of standard guidelines in pediatric and newborn care [45]. Care networks in Brazil strengthen both systems for primary care and specific priority health areas [46, 47]. A network of safety in Nepal's central purpose is to improve maternal and neonatal survival in underserved rural parts of the country [13].

**Network stakeholders.** Many different stakeholders participate in networks; this includes a range of clinical providers, health systems administrators and managers, government members, professional associations, partner organizations, and communities. Networks can be a unique opportunity for stakeholders enabling them to "work across institutional, professional, and geographic boundaries to identify and address common priorities and develop collaborative solutions" [19].

Clinicians are reported to be key in establishing and running networks, such as clinical networks in Australia [12]. They often play a hybrid or boundary spanning role between clinical and management responsibilities as in the CIN in Kenya [42] or act as a network co-chair in partnership with a health system manager, like in clinical networks in Australia [12]. Certain studies note that it is important to have a clinician in a leadership role and "influential and passionate clinical leaders" are necessary to build effective networks [48]. S4 Appendix contains reported stakeholders linked to references from the literature.

## Synthesis of results

Based on the analysis of the selected literature, networks in health systems can generally be characterized by five components: form and structure, governance and leadership, mode of

**Table 4. Reported network uses by category with corresponding number of references from the selected literature.**

| Improving and providing quality care and services | Improving patient use and experience |
|---|---|
| provide / improve / expand care / services (31) | increase use of care (11) |
| improve / provide quality and efficiency of care (16) | foster flow of knowledge and share best practices among providers and organizations and caregivers (8) |
| to change aspects of the health system / service delivery (14) | generate consumer demand (2) |
| increase access to care (12) | **Improving the health system** |
| overcome obstacles / gaps to provision of basic services / interventions (9) | to change aspects of the health system (14) |
| respond to specific health problems (9) | foster partnerships / teamwork / linkages (7) |
| promote standard / evidence-based approaches to care (8) | facilitate / improve referral (7) |
| implementation of QI initiatives (8) | collaboration (6) |
| facilitate / improve referral (6) | to facilitate coordination and cooperation (6) |
| to meet people's needs (6) | platform for problem solving (5) |
| manage patients / care (6) | studies / research / generate evidence (5) |
| collaboration (6) | develop / implement / improve models of care / UHC (5) |
| platform for problem solving (6) | coordinate complex care / continuum of care (5) |
| coordinate complex care / continuum of care (5) | shift culture of the network (3) |
| introduce new technologies / innovations (5) | use data and evidence to guide decision making (3) |
| decentralization of care / distribution of cases (2) | improve governance (3) |
| streamline patient pathways (1) | decentralization of care / distribution of cases (2) |
| Consultation (1) | streamline patient pathways (1) |
| deliver care benefits (1) | improve documentation (1) |
| **Improving provider capacity** | |
| uptake of standard guidelines / change clinical practice (16) | |
| to build capacity / skills (9) | |
| foster flow of knowledge and share best practices among providers and organizations and care givers (8) | |
| foster partnerships / teamwork / linkages (7) | |
| shift staff attitudes (1) | |

functioning, resources, and communication. The framework components and illustrative practical characteristics aim to make the network concept more concrete. These framework components create a foundation for the network that supports the network's ability to work towards achieving its purpose. The components manifest differently depending on the network context and purpose. Network contextual differences were accounted for to a certain extent by summarizing the data to a higher level of abstraction through the network components. Fig 5 shows the proposed network framework and most frequent practical characteristics.

## Discussion

### Summary of findings

This scoping review systematically searched peer-reviewed and grey literature on networks in health systems. Based on 129 pieces of selected literature, the review proposes a framework that broadly characterizes the components of networks and maps practical characteristics of

**Table 5. Reported network purposes by category with corresponding number of references from the selected literature.**

| Improving and providing quality care and services | Improving provider capacity |
|---|---|
| provide optimal / high / improve quality care (52) | to foster practice change / uptake evidence-based practices (10) |
| to transform / improve delivery of services (17) | provide information / teaching resources to providers (3) |
| address access to care (17) | provider feedback (1) |
| address challenges to providing care (8) | **Improving patient use and experience** |
| reduce / improve referral (6) | improve participation / information / experience to patients / clients (10) |
| contain / reduce healthcare costs (6) | increase uptake/use of services (3) |
| coordinate care (4) | **Improving outcomes** |
| provide integrated patient / family centered care (4) | improve outcomes (46) |
| improve coverage of services (3) | reduce hospital admissions / stays (3) |
| **Improving the health system** | reduce incidence of complication (2) |
| to transform / improve delivery of the system (17) | |
| reduce / improve referral (6) | |
| contain / reduce healthcare costs (6) | |
| linking of stakeholders / network entities (6) | |
| coordinate care (4) | |
| provide integrated patient / family centered care (4) | |
| improve coverage of services (3) | |
| improve generation and use of health data (2) | |

these components, maps the uses and purposes networks are employed to do in health systems, and identifies common stakeholders involved in networks with a focus on LMIC health systems.

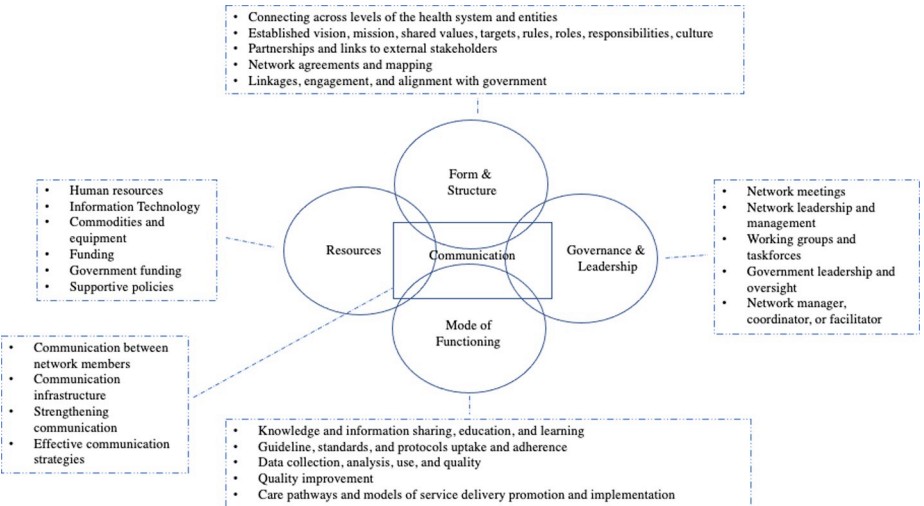

**Fig 5. Proposed network framework: Components and key practical characteristics.**

The network framework is based on diverse network examples that span geographic and clinical focus. Corresponding practical characteristics for each component illustrate how networks are operationalized in practice. While there are many different network definitions and types, the framework and illustrative practical characteristics aim to make more concrete what a network is made up of.

The network framework components are interlinked. How a network is structured can have implications for how it is governed and led, for example, a mandated or top-down network is more likely to have a command-and-control form of governance, compared to a network developed by network members [22]. Network governance and leadership are important to ensure that a network works towards achieving its goals [6, 48] and reflects how its management functions. A network cannot function without adequate resources; this includes human resources, financial resources, infrastructure, and policy [10, 48]. The types of resources required depend on the network. Networks formed from the bottom-up are often started without financial resources [15, 49], though may require resources to grow. Effective communication is needed to share the network vision and principles to stakeholders [6] and is required for the sharing and learning that is key to sustaining them [15, 41]. Communication effectuates the four other components, without which the network may be at risk of sub-optimal functioning.

The mapping of common networks uses, purposes, and stakeholders provide initial insight into why and when a network would be a useful approach in a health system and the key stakeholders involved. While uses and purposes of networks are specific to their context, the ability to summarize and identify commonalities, implies that there are underlying reasons for taking a network approach. Many different stakeholders are involved in networks; they play and contribute to the network in various roles and responsibilities, depending on the context, purpose, and functioning of the network.

Health systems are complex systems and networks have become a more frequent approach to solving problems, often complex themselves, in health systems, particularly in LMICs. Understanding what makes up a network and why a network is established, can support network implementation, and thereby enable a more considered use of network resources, leading hopefully to more rapid clinical and public health gains.

## Comparison with existing literature

This scoping review differs from other reviews and studies on health system networks because it looks across a diverse range of network examples in the literature, with a specific focus on LMICs. Previous scoping and systematic reviews focused on one type of specifically defined network: clinical networks, networks of care, quality improvement collaboratives, and clinical social franchise networks [10, 11, 21, 50–52]. A scoping study on networks of care developed a definition and evidence base for this service delivery focused concept based on LMIC and HIC literature [21]. Four systematic reviews determined the effectiveness of clinical networks in HICs, clinical social franchise networks in LMICs, and quality improvement collaboratives in both HICs and LMICs [10, 11, 50, 52]. This scoping review mapped the literature to describe a network, its uses and purposes, and the key stakeholders involved to provide a common approach for thinking about different types of networks in health systems. The review aims to go beyond explanations or results from specific types of networks to identify common features present across them and illustrate how these features happen in practice as well as to summarize the main reasons networks form and who partakes. It acknowledges that there are many diverse networks in the literature but tries to show that it is possible to identify what they share in common.

Compared to the frameworks and typologies identified in the literature and summarized in S3 Table, this paper's proposed framework is more applicable to diverse health system networks because of the more comprehensive way it captures the core elements of a network. This is because the framework is based on a comprehensive search, eliciting a significant amount of data, which was synthesized to produce the framework. All but one of the identified frameworks and typologies was HIC focused; the selected literature in this scoping review included mostly LMICs literature and therefore this proposed framework maybe more relevant to networks in these settings.

This scoping review highlights gaps in the literature on networks in health systems. Few studies look across network examples and so there are limited studies comparing different types of networks. While this scoping review's searches were focused on health system networks in LMICs, there is relatively limited literature in these geographies and so there is a need for more reporting on networks in LMIC health systems. Furthermore, much of the HIC and LMIC literature focuses on reporting clinical outcomes and less so on explaining what the network is, how it was established, how it functions and what people did; these elements are important for understanding how a network works and should be carefully characterized when establishing a network and reporting on its uses and effects. Both peer-reviewed and grey literature should consider incorporating these aspects in future publications.

There are opportunities for future research on health system networks. One area are independent evaluations of network impact, at either clinical, relational, or health system level. There were few studies that looked at measuring the success or effectiveness of networks: five systematic reviews [10, 11, 50–52], two qualitative studies [6, 48], and one cross-sectional study [12] and so measuring network effectiveness or success is another area where additional research is needed. Thirdly, improving the clarity of what networks are trying to achieve and what they think they are doing by developing theories of change or program theories would contribute to a better understanding of how networks work.

## Implications of the results

As exemplified by the literature, networks seem a useful approach to tackle health system challenges. There is limited literature that prioritizes describing what the network is and how it functions, which are important to consider when developing networks. The proposed network components provide a framework from which to do so but do not address the mechanisms that led to the networks' outcomes.

This scoping review is the first conceptualization, to our knowledge, of the common elements that make up networks and first mapping of uses, purposes, and stakeholders in LMIC networks. It is a first step towards further understanding what networks are, when and what they are used for, and the purposes they intend to achieve in health systems. This review also serves as a mapping and clarification exercise to support an in-process realist review and planned realist evaluation on understanding how and why networks work to change practices.

## Strengths and limitations

This scoping review has several strengths. It systematically searched the literature identifying and mapping relevant peer-reviewed and grey literature and created a large evidence-base of health system network examples. The network framework is based on an analysis of literature across different networks, suggesting broader applicability than frameworks developed for specific network types. This scoping review builds on and expands the knowledge from previous reviews and frameworks and typologies.

There are several limitations to this review. Due to the volume of literature on networks, the search strategy was limited to LMICs. Relevant HIC literature that the authors were already aware of or that was inadvertently returned during the database searches was reviewed. HIC network examples were retained when they provided insights relevant to networks in LMICs, particularly through study methods predominately absent in the LMIC literature. However, this approach could have missed relevant HIC literature. Secondly, many types of networks were considered beyond the scope of networks in health systems, for example, we did not include networks that were linked through financing; while financial resources are an important piece of network formation and functioning, this study focuses on networks in LMICs, which may not be financially linked, and most services require patient out-of-pocket payments. There could be relevant learnings from these excluded networks. The selected literature was limited to English and French due to reviewer capabilities, potentially excluding relevant literature in other languages, particularly literature from Latin America in Portuguese and Spanish. While we searched grey literature to try to identify examples of networks not in the peer-review literature, this search was not exhaustive and there may have been examples we missed, particularly from networks at lower levels of the health system. Furthermore, there is no standard way to report on networks. Therefore, there was significant variability in breadth and depth of information in the literature; may examples did not describe in detail how or what they did. During the evidence selection process, this led to excluding potentially relevant examples of networks due to insufficient information reported. Considering that this scoping review selected 129 pieces of peer-reviewed and grey literature, these limitations should have been mitigated to some extent as the charted data was judged sufficient to answer the research question and information saturation was reached.

In the literature, many networks are reported with similar names, though there may be considerable variability among them. This made the development of a network typology unfeasible, and the analysis of the charted data was done at a higher level of abstraction than planned. Additionally, based on how networks are reported in the literature, it was difficult to untangle the network uses and purposes as intended. Despite these limitations, the network framework and mapping were able to provide an answer to the review question.

## Conclusion

The network framework and mapping of network uses, purposes, and stakeholders has several potential applications. Firstly, the framework could be used to guide design and reporting on networks, enabling more standardized and robust information. This approach to thinking about networks could help to further understand the essential elements of a network and why it would be pertinent to achieving clinical, public health, and health system goals, which may be particularly useful as networks become a more frequent approach in strengthening health systems, particularly in LMICs. Lastly, this scoping review's results can be a launching point for further research across or within networks, particularly research that seeks to understand their mechanisms of effect.

## Supporting information

**S1 Table. Literature eligibility criteria.**
(DOCX)

**S2 Table. Data charting instrument.**
(DOCX)

**S3 Table. Existing network frameworks and typologies identified.**
(DOCX)

**S4 Table. Most frequent practical characteristics by network component with examples from the selected literature.**
(DOCX)

**S1 Appendix. Database search strategies.**
(DOCX)

**S2 Appendix. Characteristics of sources of evidence.**
(DOCX)

**S3 Appendix. Reported network components and characteristics.**
(DOCX)

**S4 Appendix. Reported network uses, purposes, and stakeholders.**
(DOCX)

**S5 Appendix. Protocol.**
(PDF)

**S6 Appendix. PRISMA ScR.**
(DOCX)

**S7 Appendix. PRISMA ScR checklist.**
(DOCX)

## Acknowledgments

We would like to thank Peter Anto Johnson for acting as a second reviewer for the literature screening and selection process and data charting.

## Author Contributions

**Conceptualization:** Katherine Kalaris, Geoff Wong, Mike English.

**Data curation:** Katherine Kalaris.

**Formal analysis:** Katherine Kalaris.

**Methodology:** Katherine Kalaris, Geoff Wong, Mike English.

**Supervision:** Geoff Wong, Mike English.

**Visualization:** Katherine Kalaris.

**Writing – original draft:** Katherine Kalaris.

**Writing – review & editing:** Katherine Kalaris, Geoff Wong, Mike English.

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
