## [Decision Letter · Decision Letter 0]

14 Nov 2022

PGPH-D-22-01359

Understanding networks in low-and middle-income countries’ health systems: a scoping review

Dear Dr. Kalaris,

Thank you for submitting your manuscript to PLOS Global Public Health. After careful consideration, we feel that it has merit but does not fully meet PLOS Global Public Health’s publication criteria as it currently stands. Therefore, we invite you to submit a revised version of the manuscript that addresses the points raised during the review process.

EDITOR: comments -

Please consider the following suggestions for strengthening the paper impact.

Methods:

Methods are clear and rigorous. One clarification that can perhaps help, is to include why you chose the databases you chose (for example is it to ensure you cover diversity of fields or to include all health system related papers). The flow chart has a step of identifying databases - so just expanding a bit on why the databases were chosen using two to three sentences will help.

Results

The phrases under each main sub-heading such as Improving and providing quality care services seems unclear. Why were those phrases picked? Were they keywords in papers? there are general terms such as 'care' that needs some explaining. Adding a bit more explanations on what Table 4 and 5 represent, leading up to the table, beyond identifying the main sub-headings will bring clarity.

Limitations:

One optional suggestion is to perhaps to mention that not all networks (even though health related) maybe captured in academic literature - therefore may not be included in this review. Mainly because many health networks in LMICs at lower level admin levels do not necessarily get represented in academic literature.

Please submit your revised manuscript by . If you will need more time than this to complete your revisions, please reply to this message or contact the journal office at globalpubhealth@plos.org. Please include the following items when submitting your revised manuscript:

We look forward to receiving your revised manuscript.

Kind regards,

Manish Barman, MD., MSc., FRCP

Academic Editor

Journal Requirements:

1. Please send a completed 'Competing Interests' statement, including any COIs declared by your co-authors. If you have no competing interests to declare, please state "The authors have declared that no competing interests exist". Otherwise please declare all competing interests beginning with the statement "I have read the journal's policy and the authors of this manuscript have the following competing interests:"

a. Please clarify all sources of funding (financial or material support) for your study. List the grants (with grant number) or organizations (with url) that supported your study, including funding received from your institution. 

b. State the initials, alongside each funding source, of each author to receive each grant.

c. State what role the funders took in the study. If the funders had no role in your study, please state: “The funders had no role in study design, data collection and analysis, decision to publish, or preparation of the manuscript.”

d. If any authors received a salary from any of your funders, please state which authors and which funders.

3. Please provide separate figure files in .tif or .eps format.

4. In the online submission form, you indicated that your data will be submitted to a repository upon acceptance.  We strongly recommend all authors deposit their data before acceptance, as the process can be lengthy and hold up publication timelines. Please note that, though access restrictions are acceptable now, your entire data will need to be made freely accessible if your manuscript is accepted for publication. This policy applies to all data except where public deposition would breach compliance with the protocol approved by your research ethics board. If you are unable to adhere to our open data policy, please kindly revise your statement to explain your reasoning and we will seek the editor's input on an exemption. Please be assured that, once you have provided your new statement, the assessment of your exemption will not hold up the peer review process.

Additional Editor Comments (if provided):

Dear Authors

Thank you for considering PLOS GPH journal for publication of your work.

The article is very well written and provides a deep insight into quality improvement by understanding the very healthcare systems which are designed to provide this care.

After careful review we have reached a decision that the manuscript can be accepted for publication with minor corrections/revision.

Thank you

Manish Barman
---

## [Editor Report · Decision Letter 1]

8 Dec 2022

Understanding networks in low-and middle-income countries’ health systems: a scoping review

PGPH-D-22-01359R1

Dear Miss Kalaris,

We are pleased to inform you that your manuscript 'Understanding networks in low-and middle-income countries’ health systems: a scoping review' has been provisionally accepted for publication in PLOS Global Public Health.

Best regards,

Manish Barman, MD., MSc., FRCP

Academic Editor

Dear Author/s

Thank you for incorporating the suggestions in your revised version.

Best

Manish